# Transcriptomic Characterization of miRNAs in *Apis cerana* Larvae Responding to *Ascosphaera apis* Infection

**DOI:** 10.3390/genes16020156

**Published:** 2025-01-26

**Authors:** Yuxuan Song, Jianfeng Qiu, Jing Kang, Ying Chen, Ruihua Cao, Wei Wang, Mengyuan Dai, Dafu Chen, Zhongmin Fu, Rui Guo

**Affiliations:** 1College of Bee Science and Biomedicine, Fujian Agriculture and Forestry University, Fuzhou 350002, China; 15757057565@163.com (Y.S.); jfqiu@fafu.edu.cn (J.Q.); kj000213@163.com (J.K.); 13979562297@163.com (Y.C.); caoruihua0103@163.com (R.C.); 13950592336@163.com (W.W.); 18050348332@163.com (M.D.); dfchen826@fafu.edu.cn (D.C.); zmf@fafu.edu.cn (Z.F.); 2National & Local United Engineering Laboratory of Natural Biotoxin, Fuzhou 350002, China; 3Apitherapy Research Institute, Fujian Agriculture and Forestry University, Fuzhou 350002, China

**Keywords:** *Ascosphaera apis*, *Apis cerana*, miRNA, larvae, immune response, transcriptome

## Abstract

*Ascosphaera apis* is a fungal pathogen that specifically infects bee larvae, causing an outbreak of chalkbrood disease in the bee colony and a decline in the number of bee colonies. The role of miRNA regulation in honeybees in response to *A. apis* infection is unclear. In this study, based on small RNA-seq, we identified the differentially expressed miRNAs (DEmiRNAs) and their regulatory networks and functions in the gut of *Apis cerana cerana* on the first day (AcT1), the second day (AcT2) and the third day (AcT3) after *A. apis* infection, and analyzed the immune response mechanism of *A. apis* through the miRNAs-mRNA regulation network of *A. apis* infection. A total of 537 miRNAs were obtained, and 10, 27, and 54 DEmiRNAs were screened in the AcT1, AcT2, and AcT3 groups, respectively. The number of DEmiRNAs gradually increased with the infection time. Stem-loop RT-PCR results showed that most of the DEmiRNAs were truly expressed, and the expression trend of DEmiRNAs was consistent with the results of sRNA-seq. The top five GO terms of DEmiRNA-targeted mRNA were binding, cellular process, catalytic activity, metabolic process, and single-organism process. The main pathways enriched by KEGG were endocytosis, ubiquitin-mediated proteolysis, phagosome, and the JAK-STAT immune-related signaling pathways. The number of DEmiRNAs and target mRNAs of these related pathway genes increased with infection time. The miRNA-mRNA regulatory network analysis showed that ace-miR-539-y was the core miRNA of the early immune response in the gut of larvae infected with *A. apis* in the JAK-STAT pathway and phagosome, and ace-miR-1277-x was the core miRNA of the late immune response in the gut of larvae infected with *A. apis* in the JAK-STAT signaling pathway and phagosome. The results showed that miRNA participated in the immune response of honeybees to *A. apis* infection by regulating the host’s energy metabolism, cellular immunity, and humoral immunity. The results of this study provide a basis for the regulation of miRNAs in *A. c. cerana* larvae in response to *A. apis* infection and provide new insights into host-pathogen interactions.

## 1. Introduction

*A. c. cerana* is a subspecies of *A. cerana*. It is also a unique local bee species in China and one of the main bee species used in beekeeping production, and after long-term evolution, it has been highly adapted to the Chinese geographical environment [1]. *A. apis* is a fungal pathogen that specifically infects bee larvae. It has led to the outbreak of chalkbrood disease in colonies, a sharp decrease in the number of larvae in bee colonies, and a decline in bee population, causing huge economic losses to beekeepers [2].

The miRNAs are a class of small non-coding RNAs with a length of 18–25 nt that regulate gene expression at the post-transcriptional level, thereby affecting many biological processes, such as immune defense and development [3]. In recent years, with the rapid development of second-generation sequencing technology and bioinformatics, the transcriptomics and non-coding RNA omics of model organisms such as *Mus musculus* [4], *Arabidopsis thaliana* [5], and *Saccharomyces cerevisiae* [6] have been studied more and more deeply, and a large number of miRNAs have been predicted and identified. However, for most insects except *Drosophila* and *Bombyx mori*, the related research is still in its infancy. Previously, the extensive research on miRNA has found that it plays an important regulatory role in the growth and development, behavior and immune defense of insects [7]. For example, Zhang [8] reported that four new microRNAs regulate gene expression of *Manduca sexta* for pattern recognition, prophenoloxidase activation, cellular responses, antimicrobial peptide synthesis, and conserved intracellular signal transduction. Karamipour et al. [9] discovered that the microRNA pathway is involved in *Spodoptera frugiperda* (Sf9) cells’ antiviral immune defense against *Autographa californica* multiple nucleopolyhedrovirus infection, and the results suggest the antiviral role of the miRNA pathway in defending Sf9 cells against *AcMNPV*. The previous studies have shown that miRNAs play an important role in the regulation of memory formation, caste differentiation, and immune defense in honeybees [10,11,12]. Studies have shown that amc-miR-34 plays a regulatory role in the immune response of *Apis mellifera* to *A. apis* infection by positively regulating the expression of *hsp* and *abct* [13]. Studies have shown that viruses or parasites produce host-like miRNAs that regulate host gene expression [14]. Over the past decade, miRNAs of *Leishmania* (*Leishmania donovani*) have been shown to be involved in the pathogenesis of leishmaniasis [15]. miR-361-3p and miR-140-3p were significantly overexpressed in cutaneous leishmaniasis lesions generated by *L. braziliensis* infection as compared to normal skin samples, targeting genes involved in the worsening of tissue damage [16]. By analyzing the expression profiles of host and pathogen miRNAs and their target genes during honeybee infections, Evans et al. [17] found that 918 honeybee genes involved in biological processes such as apoptosis and the innate immune response were broadly negatively regulated by five parasite miRNAs. Fan et al. [18] found that a total of 343 down-regulated mRNAs in *A. mellifera* were putative targets for 121 up-regulated miRNAs in *Nosema cerana*, which were mainly enriched in 217 KEGG pathways, including the JAK-STAT signaling pathway, endocytosis, and lysosomes. However, the current research on differentially expressed miRNAs in *A. c. cerana* in response to Glomus infection is generally limited and not deep enough.

In the early stage, our research group conducted a detailed study on the immune response mechanism of *Apis mellifera ligustica* and *A. c. cerana* larvae under the stress of *A. apis* in transcriptomics and non-coding RNA and deeply analyzed the regulatory relationship between circular RNA, long-chain non-coding RNA, miRNA, and mRNA [19,20]. In this study, based on bioinformatics methods, the important regulatory effect of miRNA on bee immunity in the intestinal tract of *A. c. cerana* larvae in response to Glomus stress was explained at the miRNA level.

## 2. Materials and Methods

### 2.1. Honey Bee Larvae and Fungi

*A. c. cerana* worker larvae were derived from three healthy brood combs of three different strong colonies (The new queen lays about 800 eggs a day, the number of worker bees fills the four nest frames) reared in the apiary of the College of Bee Science and Biomedicine, Fujian Agriculture and Forestry University, Fuzhou, China. The chalkbrood mummies were taken from colonies reared in the apiary of the College of Bee Science and Biomedicine, Fujian Agriculture and Forestry University, Fuzhou, China.

We performed PCR validation on *A. c. cerana* worker samples from colonies to ensure that the bee larvae were not infected with *A. apis*. The intestines of 20 larvae were dissected with tweezers sterilized with 70% alcohol and placed in sterile tubes (*n* = 20). The total RNA of the intestinal tracts of bee larvae was extracted with the gRNAiso plus kit (TaKaRa, Dalian, China), and then reverse transcribed into cDNA. We used the RT-PCR system (20 μL): cDNA 1 μL, 2 × Hieff^®^PCR Master Mix 10 μL (Yeasen, Shanghai, China), Primer-F (5′-TCTGGCGGCCGGTTAAAGGCTTC-3′) 1 μL, Primer-R (5′-GTTTCAAGACGGGCCACAAAC-3′) 1 μL, ddH2O 7 μL. We used the following PCR procedure: pre-denaturation at 95 °C for 5 min; denaturation at 95 °C for 40 s, annealing at 55 °C for 30 s, extension at 72 °C for 40 s, 34 cycles; extending at 72 °C for 10 min. The products were detected using 1.5% agarose gel electrophoresis and a gel imager (Peiqing, China). Colonies of worker bee samples without amplification bands were selected for future research. Based on the method described by Jensen [21], a fresh chalkbrood mummy was sterilized using 10% sodium hypochlorite for 10 min and then rinsed in sterile distilled water for 2 min. Subsequently, the mummy was cut into smaller pieces and cultured at 25 °C on plates of potato dextrose agar (PDA) medium, and at 7 days after culturing, fresh spores of *A. apis* were purified.

### 2.2. Preparation of Larval Gut Samples

Frames of sealed brood comb from a healthy colony of *A. c. cerana* (*A. apis*-free, as verified by PCR) kept in an experimental apiary of the College of Bee Science and Biomedicine, Fujian Agriculture and Forestry University were swiftly transferred to the laboratory and kept in an incubator at 34 ± 2 °C to provide newly emerged *A. apis*-free workers. The 2-day-old larvae (*n* = 96) were transplanted into two sterile 48-well cell culture plate with preset 40 μL feed (preheated at 35 °C) with a shifting needle, and cultured in a constant temperature and humidity box at 35 ± 0.5 °C and 90% relative humidity. The feed was changed every 24 h. Larvae in the treatment group (*n* = 48) were fed a premixed diet containing 20 μL of conidia at 3 days of age, with a conidia concentration of 1 × 10^7^ spores/mL. The 3-day-old larvae in the control group (*n* = 48) were fed a diet without conidia 20 μL with a 70% alcohol disinfected shifting needle, and then all larvae were fed a diet without conidia on the ultra-clean workbench. The control group and *A. apis* infected group were reared in different sterile incubators to avoid potential infection. The guts of larvae in the control group (AcCK) and the *A. apis*-infected group (AcT) were collected on day 1, day 2, and day 3 after infection, respectively. Three gut samples were mixed as one biological sample, and the experiment used three independent biological samples. The AcCK and AcT group are three biological repeats. The 48-well cell culture plate was removed from the incubator with constant temperature and humidity and placed on the ultra-clean workbench. The medium bee larvae were gently picked up with tweezers and washed in sterile water to remove the feeding liquid on the insect’s body surface, then placed on the anatomical plate for dissection, during which the excess fat body around the intestine was removed. The dissected larval intestines were placed in one 1.5 mL RNA-free EP tube for every 9 larvae, which were quickly transferred to the ultra-low temperature refrigerator at −80 °C for storage.

### 2.3. Small RNA Sequencing (sRNA-seq) and Quality Control

First, the total RNA of each gut sample in the *A. apis*-infected and control groups was extracted using TRIzol Reagent (Invitrogen, Carlsbad, CA, USA) following the manufacturer’s protocols. Second, DNA contaminants were removed with RNase-free DNase I (TaKaRa, Beijing, China). The purified RNA quantity and quality were checked using a Nanodrop 2000 spectrophotometer (Thermo Fisher, Waltham, MA, USA), and the integrity of the RNA samples was evaluated using an Agilent 2100 bioanalyzer (Agilent Technologies, Santa Clara, CA, USA). Only values of 28S/18S ≥ 0.7 and RIN ≥ 7.0 were considered qualified for the subsequent small RNA library construction. Thirdly, RNA molecules in the size range of 18–30 nt were enriched by agarose gel electrophoresis (AGE) and then ligated with 30 and 50 RNA adaptors. Fragments with adaptors on both ends were enriched by PCR after reverse transcription. Fourth, the subsequent cDNAs were purified and enriched by 3.5% AGE to isolate the expected size (140–160 bp) fractions and eliminate unincorporated primers, primer dimer products, and dimerized adaptors. Ultimately, the 18 cDNA libraries were sequenced on the Illumina sequencing platform (HiSeqTM 4000) using the single-end technology made by GENE DENOVO Biotechnology Co. (Guangzhou, China). The names of these sequencing samples are shown in Table 1. The data measured in this study have been uploaded to the National Center for Biotechnology Information (NCBISRA) database, BioProject number: PRJNA395108.

The original sequencing image data were transferred into sequence data via base calling, which is defined as raw data or raw reads stored in the FASTA format. Then, the raw reads of all samples were pre-processed by removing adaptor sequences and reads with more than 5% unknown nucleotides. Low-quality reads, defined as reads where the percentage of low-quality bases of quality (Q) value 5 was more than 50% in a read, were also removed. The clean reads were aligned to the *A. mellifera* genome assembly Amel_4.5 using SOAP aligner/soap2, with the threshold that no more than two mismatches were permitted in the alignment.

### 2.4. Quality Control of Raw Data

The RNA fragments of 18–30 nt were selected for agarose gel electrophoresis, and then 3′ RNA adapters were ligated. The connected products were separated by 15% denatured PAGE gel electrophoresis, and the target bands of 36–44 nt were selected for gel cutting. The gelling products were recovered, the 5′ RNA adapters were ligated, and then the sRNA samples connected with the bilateral joints were performed by reverse transcription PCR. The reverse-transcribed product was separated by 3.5% agarose gel electrophoresis, and a band of 140–160 bp region was selected for gelling. The final library was the product of the gelling recovery and built library computer sequencing. Original data quality control conditions: (1) Low-quality reads (reads sequences with a base identification error rate greater than 1% or containing unknown base N) were filtered from the original data; (2) Reads without the 3′ connector were filtered out and sequences before the 3′ connector were intercepted; (3) Filter the 5′ connector, and filter the reads after the connector that are short, contain poly A, or do not contain inserted fragments. From this process, we got high quality sequence tags (clean tags).

### 2.5. Identification of miRNAs

Blastall software (version 2.2.25) was used to annotate the obtained clean tags in GenBank and Rfam databases. We filtered to remove ribosome RNA (rRNA), small cytosol RNA (scRNA), small nucleolar RNA (snoRNA,) an intracellular small nuclear RNA (snRNA). and transfer RNA (tRNA). The Bowite (version 2.1.2) and RepeatMasker (version 4.0.6) software packages were used to compare the clean tags’ reference genome (assemblyACSNU-2.0) and repeat sequences. Its position on the genome was determined and the small RNA tags sequence repeat associated was found. After filtering and matching the sequence tags to the mapped tags, we completed the miRBase database annotation.

### 2.6. Screening of DEmiRNAs

Bowite software (version 2.5.4) was used to compare the known miRNA sequences in the miRBase database with the mapped tags to obtain the expression of known miRNA and the possible precursor sequences. The expression level of miRNA in each sample was calculated, and the expression level was normalized by a TPM (transcripts per million) algorithm formula (TPM = T × 10^6^/N, T represents miRNA tags, N represents total miRNA tags) and the expression profile of all miRNAs in the sample was obtained. Then, based on the method: (mirRNA’s RPM in AcCK1)/(mirRNA’s RPM in AcT1), (mirRNA’s RPM in AcCK2)/(mirRNA’s RPM in AcT2), and (mirRNA’s RPM in AcCK3)/(mirRNA’s RPM in AcT3), the Fold Change (FC) between AcCK1 vs. AcT1, AcCK2 vs. AcT2, and AcCK3 vs. AcT3 groups was computed. The selected DEmiRNA filter condition was |log_2_(Fold change)| ≥ 1, *p* ≤ 0.05.

### 2.7. Stem-Loop RT-PCR Verification of miRNAs

The DEmiRNA of 10, 15, and 25 medium *A. c. cerana* were randomly selected for stem-loop RT-PCR identification. The total RNA from the intestinal tracts of bee larvae was extracted by a g RNAiso plus kit (TaKaRa), and followed by inverse transcription with corresponding stem-loop primers using the method mentioned above. The generated cDNA was used for RT-qPCR validation with a specific forward primer and universal reverse primer. RT-qPCR was carried out in an Applied Biosystems QuantStudio 3 (Thermo Fisher, Waltham, MA, USA). The resulting cDNA serves as a template for RT-PCR. The RT-PCR system was then set up as described above in Section 2.1. The primers used in this study are shown in Table 2.

### 2.8. Target Prediction of DEmiRNAs

The target mRNAs for the miRNAs were predicted with miRanda (v3.3a) [22], RNAhybrid (v2.1.2) +svm_light (v6.01) [23], and TargetFinder (Version: 7.0) [24] software. The input files were miRNA FASTA sequences files. Intersections of the results from the three above mentioned programs comprised the final predicted gene targets. Then, miRNA-mRNA regulation networks were visualized using Cytoscape v.3.2.1 software [25] using the following parameters: the free energy of the target site/miRNA duplex needs to be lower than −35 kcal/mol and *p* < 0.05.

### 2.9. Stem-Loop RT-qPCR Detection of DEmiRNAs

Four medium *A. c. cerana* DEmiRNAs were randomly selected in each group for stem-loop RT-qPCR identification. The total RNA of bee larvae intestinal samples was extracted using an RNA extraction kit (TaKaRa, Dalian, China) and stem ring primers were used to synthesize the total RNA of extracted samples for the first cDNA strand. The resulting cDNA was used as an RT-qPCR template. The RT-qPCR system used was 20 μL: SYBR Green Pre-mix (Vazyme, Nanjing, China) was 10 μL; The primers and cDNA template were 1 μL and 2 μL, respectively; DEPC water to 20 μL (primers see Table 1). According to the instructions of the RT-qPCR kit, the annealing temperature was 52 °C. The RT-qPCR products were imaged using a gel imager after agarose gel electrophoresis. U6 was the internal reference gene, and the relative expression of miRNA was analyzed by a 2^−ΔΔCt^ algorithm. Three biological and technical replicates were made for each group of samples, and the experimental data were drawn after analysis.

## 3. Results

### 3.1. Characterization of miRNAs in the A. c. cerana Larval Guts

A total of 537 miRNAs were predicted in this study, and the lengths of these miRNAs were mainly distributed between 18 and 26 nt, with 22 nt and 23 nt accounting for the largest proportion of 31.31% (Figure 1A). Predicted from the AcCK1 group were 248, 277, and 255 miRNAs; 259, 279, and 277 miRNAs were predicted from the AcT1 groups (Figure 1B). 289, 253, and 272 miRNAs were predicted from the AcCK2 groups; 32, 249, and 247 miRNAs were predicted from the AcT5 groups (Figure 1C); 247, 241, and 258 miRNAs were predicted from the AcCK3 groups; 217, 229, 216 miRNAs were predicted from the AcT3 groups (Figure 1D). Moreover, after the analysis of the base bias of total miRNA at each position, the results demonstrated that the first base of miRNA was mainly the U base (Figure 2), and the proportion of each base of the total miRNA was similar, but the proportions of the U, A, G and C bases in the same position and each position was significantly different (Figure 3).

A total of 73 DEmiRNAs were obtained. There were 5 miRNAs up-regulated and 5 miRNAs down-regulated in the AcCK1 vs. AcCT1 group, respectively. Seventeen miRNAs were up-regulated and 10 miRNAs were down-regulated in the AcCK2 vs. AcCT2 group. In the AcCK3 vs. AcCT3 group, 31 miRNAs were up-regulated and 23 miRNAs were down-regulated (Figure 4A). Venn analysis showed that only two DEmiRNAs were shared in the guts of workers of different ages and between groups. There were 8, 11, and 38 unique DEmiRNAs in the AcCK1 vs. AcCT1 group, AcCK2 vs. AcCT2 group, and AcCK3 vs. AcCT3 group, respectively (Figure 4B).

### 3.2. Stem-Loop RT-PCR Identification of DEmiRNA

Stem-loop RT-PCR showed that 10 DEmiRNAs randomly selected from the guts of the AcCK1 vs. AcT1 groups could amplify DNA bands (Figure 5A). Five DEmiRNAs found in the guts of the AcCK2 vs. AcT2 groups could be successfully amplified by stem-loop RT-PCR (Figure 5B). Ten DEmiRNAs found in the guts of the AcCK3 vs. AcT3 groups could be successfully amplified (Figure 5C). The above results indicate that most of the selected DEmiRNAs were authentic.

The four DEmiRNAs were randomly selected from each AcCK and AcT group to verify whether their expression trends were consistent with the sRNA-seq results. The stem-loop RT-qPCR showed that the amplified miRNA were all single bands and the expression trend of DEmiRNA was consistent with the sequencing data, except for ace-novel-m0029-5p and ace-miR-352-y, which were contrary to the sequencing data (Figure 6). The stem-loop RT-qPCR indicated that the results of sRNA-seq were accurate.

### 3.3. Target Gene Prediction and Functional Annotation of DEmiRNAs in the Gut of A. cerana Larvae

Analysis of the regulatory network of the DEmiRNAs-mRNA showed that a total of 2778 target mRNAs was predicted by 10 DEmiRNAs of the AcCK1 vs. AcT1 group, and the top 3 DEmiRNAs with the largest number of regulatory mRNAs were ace-miR-539-y, ace-miR-152-y, and ace-novel-m0029-5p. A total of 8769 target mRNAs was predicted by 27 DEmiRNAs of the AcCK2 vs. AcT2 group. The top 3 DEmiRNAs with the largest number of regulatory mRNAs were ace-miR-1277-x, ace-novel-m0053-5p, and ace-novel-m0054-5p. A total of 9774 target mRNA was predicted by 54 DEmiRNAs of the AcCK3 vs. AcT3 group. The DEmiRNAs with the top 3 largest numbers of regulatory mRNAs were ace-miR-1277-x, ace-novel-m0024-3p, and ace-miR-6052-x (Table 3).

DEmiRNA-targeted mRNA GO classification results showed that AcCK1 vs. AcT1 group DEmiRNA target mRNA was noted on 43 GO terms of biological processes, cellular components, and molecular functions. The top 5 GO terms of target mRNA enrichment numbers were binding (406), cellular processes (318), catalytic activities (264), metabolic processes (251), and single-organism processes (225). In addition, the numbers of target mRNAs enriched on the immune system progression and stress GO terms were 79 and 2, respectively (Figure 7A). The target mRNAs of DEmiRNAs in both the AcCK2 vs. AcT2 group and the AcCK3 vs. AcT3 group were enriched to 47 GO terms. The top 5 GO terms were consistent with those enriched in the AcCK1 vs. AcT1 group. The DEmiRNAs of the AcCK2 vs. AcT2 groups and the AcCK3 vs. AcT3 group targeted the mitochondrial dehydrogenase (XM_017060577.1) and mitochondrial thioredoxin reductase (XM_017051927.1) genes and were enriched on detoxification (Figure 6B and Figure 7A). However, this was not found in the AcCK1 vs. AcT1 group. The target mRNAs of the DEmiRNAs were significantly enriched on GO terms related to signal transduction in the AcCK1 vs. AcT1 group, such as guanosine nucleotide exchange factor activity (GO: 0005088), kinase regulation activity (GO: 0019207), and receptor signaling protein activity (GO: 0005057) (Figure 7A).

The enrichment analysis of the KEGG pathway showed that 248 target mRNAs of the AcCK1 vs. AcT1 group DEmiRNA could be annotated to 108 signaling pathways, among which the functions of the top 5 enriched target mRNAs were endocytosis, ubiquitin-mediated proteolysis, the mRNA surveillance pathway, the Wnt signaling pathway, and RNA transport (Figure 7D). The 1004 target mRNAs of the AcCK2 vs. AcT2 group DEmiRNA were annotated to 146 signaling pathways (Figure 7E), and the 1117 target mRNAs of the AcCK3 vs. AcT3 group DEmiRNA could be annotated to 136 signaling pathways (Figure 7F). The functions of the top 5 target mRNAs they enriched were all endocytosis, endoplasmic reticulum protein processing, carbon metabolism, ubiquitin-mediated proteolysis, and RNA transport (Figure 7E,F).

Furthermore, some of DEmiRNAs’ target mRNAs were significantly enriched in the endocytosis, phagosome, and JAK-STAT immune-related signaling pathways (*p* < 0.05), as well as in the FoXO signaling pathways related to apoptosis. We focused on the JAK-STAT signaling pathway and the phagosome-enriched miRNA, and targeted transcripts were used to establish the regulatory network diagram. The results showed that ace-miR-152-y and ace-miR-539-y were in the central position of the AcCK1 vs. AcT1 group in the DEmiRNA-mRNA regulatory network (Figure 8A). A total of 11 miRNAs were in the center of the AcCK2 vs. AcT2 group (Figure 8B). A total of 18 miRNAs were in the center of the AcCK3 vs. AcT3 group (Figure 8C).

## 4. Discussion

Over the past decade, miRNA has become a global research hotspot and has attracted more and more attention [26]. The vast majority of miRNA research is carried out in animals, plants, and insects, especially in model species such as *Homo sapiens* [27], *A. thaliana* [28], and *Drosophila* [29]. Research into miRNA in bees has also received increasing attention. In previous studies, bumblebees (*Bombus terrestris*) infected with the slow bee paralysis virus (SBPV) showed increased expression of two miRNAs, dicer-1 and ago-1. There were 17 and 12 differentially expressed miRNAs in SBPV and Israeli acute paralysis virus (IAPV) infection, respectively [30]. Female alfalfa leafcutting bees (*Megachile rotundata*) adjust their miRNA deposition in response to seasonal changes [31]. Ame-miR-980-3p is involved in *A. mellifera* autophagy mediated midgut remodeling by targeting *Atg2B* [32]. Ame-miR-519 influences the level of juvenile hormone through *Eth*, thereby regulating the transition from larva to pupa of *A. mellifera* workers [33]. ame-miR-34 modulates larval body weight and the immune response of *A. mellifera* workers to invasion by *A. apis* [13]. The function of miRNAs is relatively conserved in both *A. c. cerana* and *A. mellifera*.

In this study, based on sRNA-seq, this study analyzed the miRNA differences in the larval intestine of *A. c. cerana* at 1 dpi, 2 dpi, and 3 dpi. The results showed that a total of 537 miRNAs were detected and 73 DEmiRNAs were obtained. DEmiRNAs targeted 2778 mRNAs, and GO entries were enriched to binding, cellular process, catalytic activity, metabolic process, and single-organism processes (Figure 7A). The main pathways enriched by KEGG were endocytosis, ubiquitin-mediated proteolysis, RNA transport, JAK-STAT immune-related signaling pathways, and phagosomes (Figure 7), and the miRNA-mRNA regulatory network enriched by the JAK-STAT signaling pathway was established (Figure 7D). The results of this study provide a rich genetic resource for the function of miRNAs in the response of *A. c. cerana* larvae to *A. apis* infection, and provide new insights into host-pathogen interactions.

After *A. apis* infection, the length of miRNA detected in the gut of *A. c. cerana* larvae was mainly distributed between 18 nt and 25 nt, and the peak was between 22 nt and 23 nt (Figure 1). The first base of miRNA was mainly the U base (Figure 2), which was consistent with the miRNA found in *Equus caballus* [34], *B. mori* [35], and *Crassostrea gigas* [36]. The number of DEmiRNAs in the gut of *A. c. cerana* larvae increased gradually from 1 dpi to 3 dpi, and the number of unique DEmiRNAs was also the highest at 3 dpi, with only two common DEmiRNAs (Figure 4). This may be due to the gradual activation of the immune response with the increase of *A. apis* infection time, resulting in the increase of DEmiRNA. The results of DEmiRNA expression verification also showed that the expression of the screened DEmiRNA was highly consistent, indicating that the sequencing data had high reliability (Figure 5 and Figure 6).

A single miRNA simultaneously regulates multiple target mRNAs to inhibit their expression, and vice versa [37]. We observed that the DEmiRNAs with the largest numbers of target genes regulated by the gut of *A. c. cerana* larvae in response to *A. apis* infection were ace-miR-539-y and ace-miR-1277-x, suggesting that ace-miR-539-y and ace-miR-1277-x may play an important regulatory role in the response of *A. c. cerana* larvae to *A. apis* infection (Figure 8). As combined with the previous studies, which found that 165 differentially expressed miRNAs of *N. ceranae* as a whole targeted 11,711 mRNAs in the midgut of *A. mellifera* workers [38], we suggest that miRNAs target multiple mRNAs, taking part in the metabolism and immunity process of two *Apis* species of workers. As an important regulator of insect immunity, miRNAs directly target immune-related genes [39]. For example, under normal non-pathogenic conditions, the conserved miR-8 in the fat body can regulate the innate immune balance of *Drosophila* by negatively regulating the expression of antimicrobial peptides [40]. Ame-miR-34 plays a regulatory role in the host’s immune response to the invasion of *A. apis* by positively regulating the expression of *hsp* and *abct* in the gut of uninfected and infected *A. apis* larvae [13]. In the process of *A. c. cerana*’s response to *N. ceranae*, miR-598 and miR-3654-y are located in the center of the DEmiRNA regulatory network, which may be a key regulator of the interaction between *A. c. cerana* and *N. ceranae* [12].

GO and KEGG enrichment of DEmiRNA target mRNA was performed to analyze the functions and pathways related to larval gut response to *A. apis* infection. The main pathways include the GO cellular process, catalytic activity, metabolic processes, KEGG endocytosis, RNA transport, phagosome, and JAK-STAT immune-related signaling pathways (Figure 7). Interestingly, the number of DEmiRNAs and target mRNAs of these related pathway genes increased with the duration of the infection (Figure 7). The results were indicative of the involvement of corresponding DEmiRNAs and target mRNAs in modulating the cellular and humoral immune responses in the gut of *A. c. cerana* during the developmental process.

The release, transfer, storage, and utilization of energy in the metabolic process play an indispensable role in insect immune response [41]. The invasion of pathogenic microorganisms not only activates cellular immunity, including endocytosis, autophagy, and phagosome [42], but also activates humoral immunity [43], such as the JAK-STAT, Toll, and Imd signaling pathways [44]. Wang et al. [45] analyzed that the 20E-HaEcR-HaUSP complex of *Helicoverpa armigera* stimulated the expression of the *HaCTL*1 gene in hemolymph, thereby increasing the synthesis and secretion of the HaCTL1 protein, promoting cell phagocytosis and encapsulation and resisting the invasion of *Ovomermis sinensis*. Lourenço et al. [46] detected that the expression of the antimicrobial peptide gene *Abaecin* in *A. mellifera* was regulated by both the Toll and IMD signaling pathways through bioinformatics analysis, and the expression of *Hymenoptaecin* and *Defensin*1 was regulated by the IMD and Toll signaling pathways, respectively. Studies have shown that the JAK-STAT signaling pathway and phagosome, as important components of the insect innate immune system, were significantly activated during pathogen infection and are widely involved in apoptosis, immune regulation, and other processes [47]. This study found that in the JAK-STAT signaling pathway, ace-miR-152-y and ace-miR-539-y were located in the center of the DEmiRNA-mRNA regulatory network on the JAK-STAT signaling pathway in the intestinal group of 1 dpi larvae, and ace-miR-539-y was also located in the center of the DEmiRNA-mRNA regulatory network on phagosome in the intestinal group of 1 dpi larvae, indicating that ace-miR-539-y may be the key miRNA in the JAK-STAT signaling pathway and phagosome during the early response of *A. c. cerana* to *A. apis* infection, respectively (Figure 8A,D). The ace-miR-1277-x is at the center of the regulatory network in the late stage of the host’s infection with *A. apis*, indicating that ace-miR-1277-x may be a key miRNA in the JAK-STAT signaling pathway and phagosome that regulates the response of bees to *A. apis* in the late stage of *A. apis* infection (Figure 8B,C,E,F). Further analysis of the KEGG pathway found that ace-miR-1277-x may affect the JAK-STAT signaling pathway by regulating mRNAs such as the E3 SUMO protein ligase PIAS3 gene (XM_017054673.1), EOS protein 2B gene (XM_017053215.1), and SOS protein gene (XM_017058556.1), thereby affecting the transmission process of the JAK-STAT signaling pathway and participating in immune response (Figure 7E,F). Previous studies have found that ace-miR-152 inhibits the proliferation and invasion of non-small cell lung cancer (NSCLC) cells by inhibiting fibroblast growth factor 2 (FGF2) [48]. As the core of the regulatory network, ace-miR-1277-x participates in a host’s immune defense by affecting apoptosis during the response of microRNA in the intestinal tract of *A. c. cerana* larvae to *A. apis* [49]. MiR-539 acts as a tumor suppressor in prostate cancer by down-regulating Distal-less 1 through the TGF-β/Smad4 signaling pathway [50].

In this study, the DEmiRNA-mRNA regulatory network was used to screen out the key miRNAs (ace-miR-539-y and ace-miR-1277-x) that regulate the JAK-STAT signaling pathway. It can be used as a candidate miRNA for further functional research.

## Figures and Tables

**Figure 1 genes-16-00156-f001:**
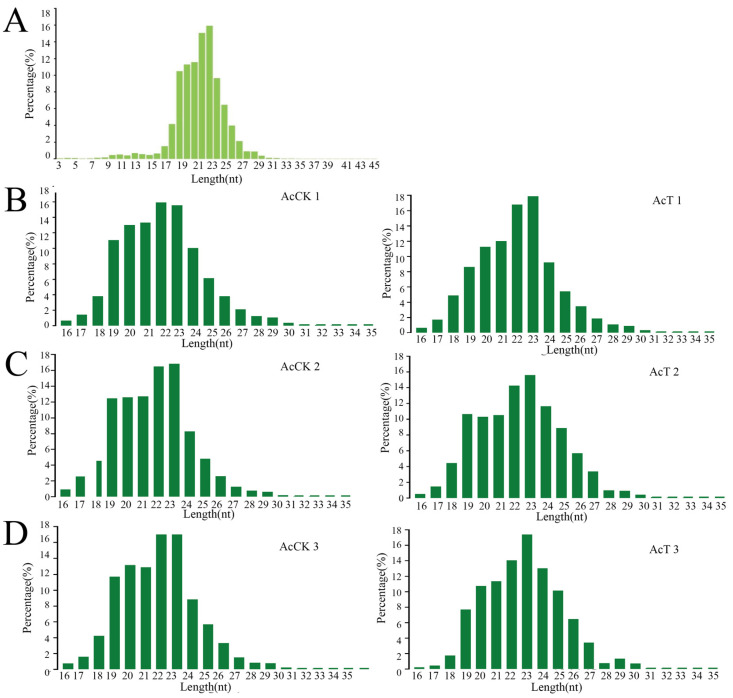
Analysis the length characteristics of miRNAs in the larval gut of *A. c. cerane.* (**A**) Total miRNAs length distribution in the gut of *A. c. cerana* larvae. (**B**) The length distribution of miRNAs in AcCK1 vs. AcT1 groups. (**C**) The length distribution of miRNAs in AcCK2 vs. AcT2 groups. (**D**) The length distribution of miRNAs in AcCK3 vs. AcT3 groups.

**Figure 2 genes-16-00156-f002:**
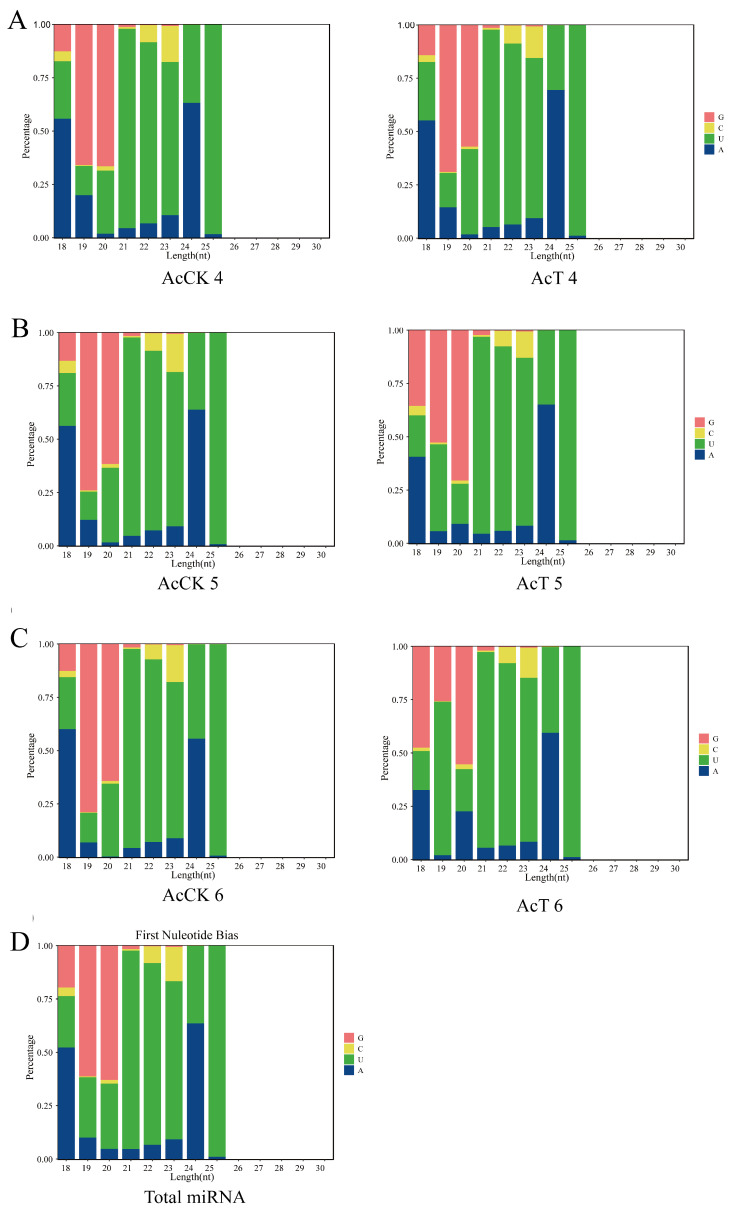
The first base bias of miRNAs. (**A**) Nucleotide bias at first base in gut of AcCK1 group and AcCT1 group. (**B**) Nucleotide bias at first base in gut of AcCK2 group and AcCT2 group. (**C**) Nucleotide bias at first base in gut of AcCK3 group and AcCT3 group. (**D**) Nucleotide bias at first base in gut of *A. c. cerana* larvae.

**Figure 3 genes-16-00156-f003:**
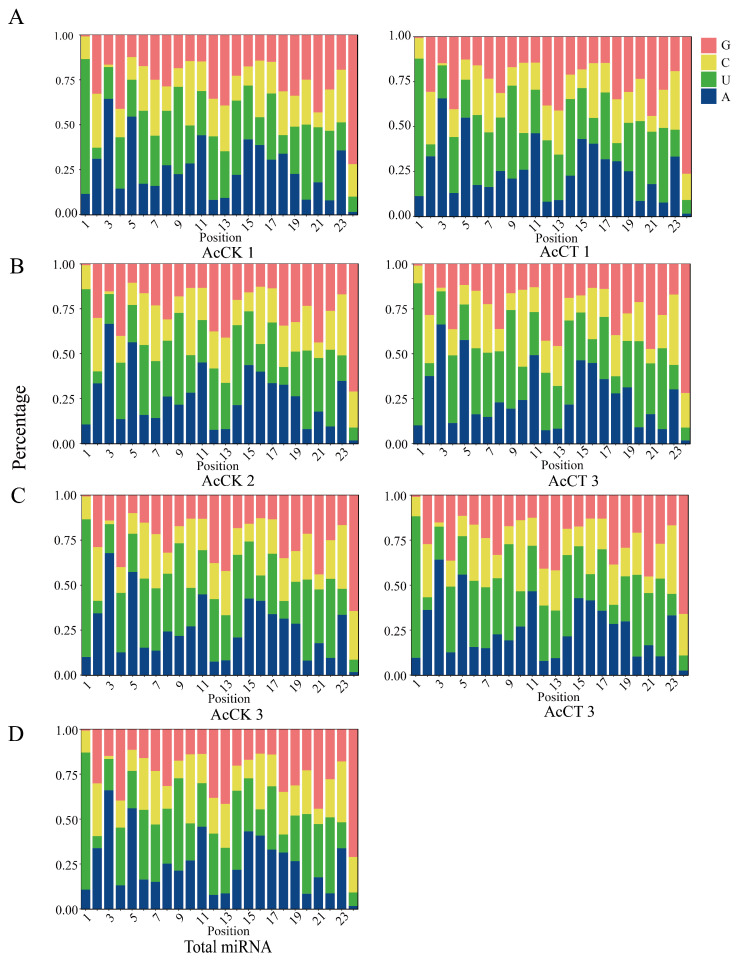
Base bias of miRNAs at each position. (**A**) Nucleotide bias at each position in guts of 4-day-old larvae. (**B**) Nucleotide bias at each position in guts of 5-day-old larvae. (**C**) Nucleotide bias at each position in guts of 6-day-old larvae. (**D**) Nucleotide bias at each position in gut of *A. c. cerana* larvae.

**Figure 4 genes-16-00156-f004:**
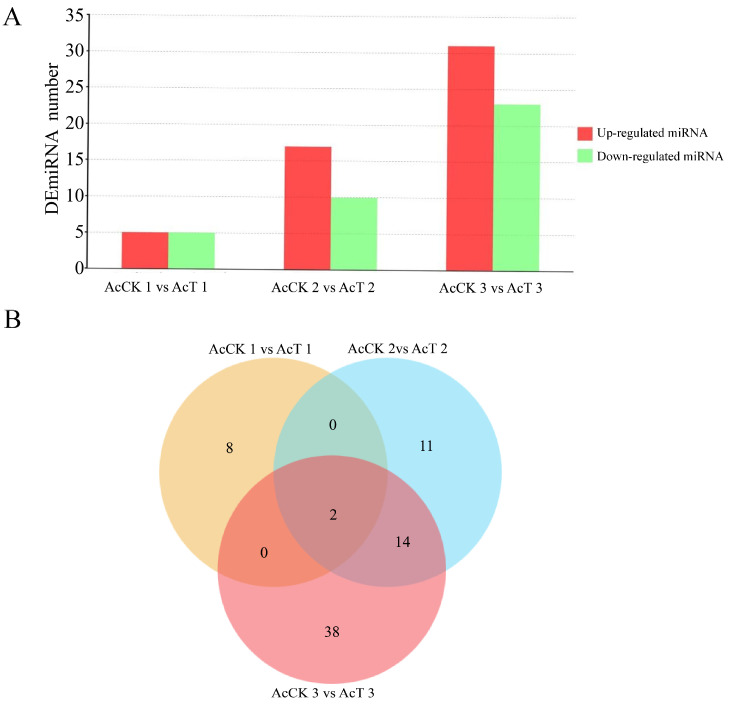
DEmiRNA analysis in AcCK vs. AcT group. (**A**) DEmiRNAs in the gut of control group and *A. apis* infection group. (**B**) Venn analysis of DEmiRNAs. The number in (B) represents the number of genes.

**Figure 5 genes-16-00156-f005:**
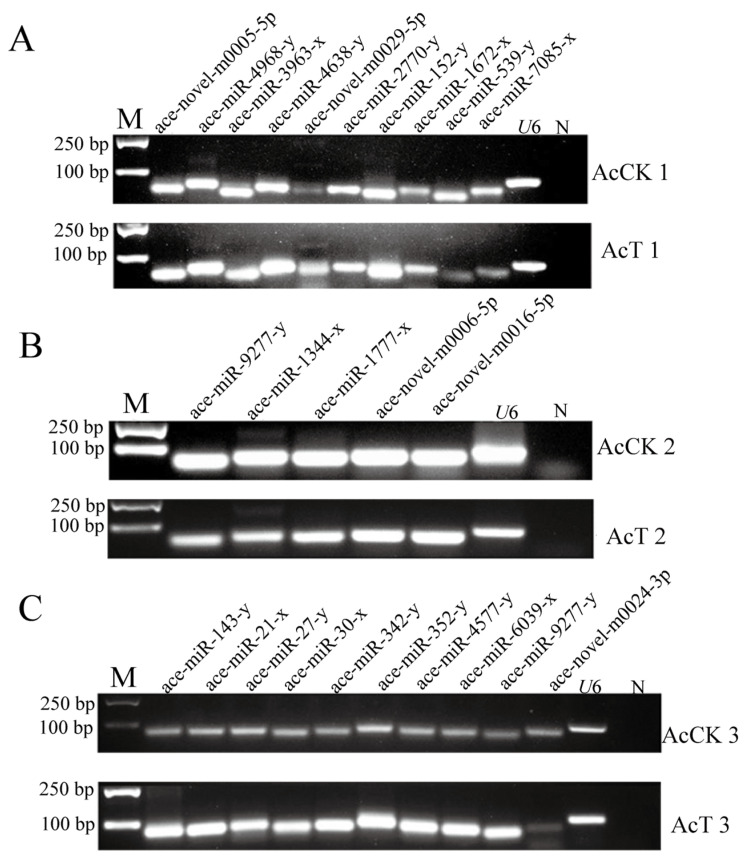
Identification of DEmiRNA by stem-loop RT-PCR. (**A**) Stem-loop RT-PCR identification of 10 DEmiRNAs in AcCK1 and AcT1 group. (**B**) Stem-loop RT-PCR identification of 5 DEmiRNAs in AcCK2 and AcT2 group. (**C**) Stem-loop RT-PCR identification of 10 DEmiRNAs in AcCK3 and AcT3 group. M: DNA marker; N: Negative control.

**Figure 6 genes-16-00156-f006:**
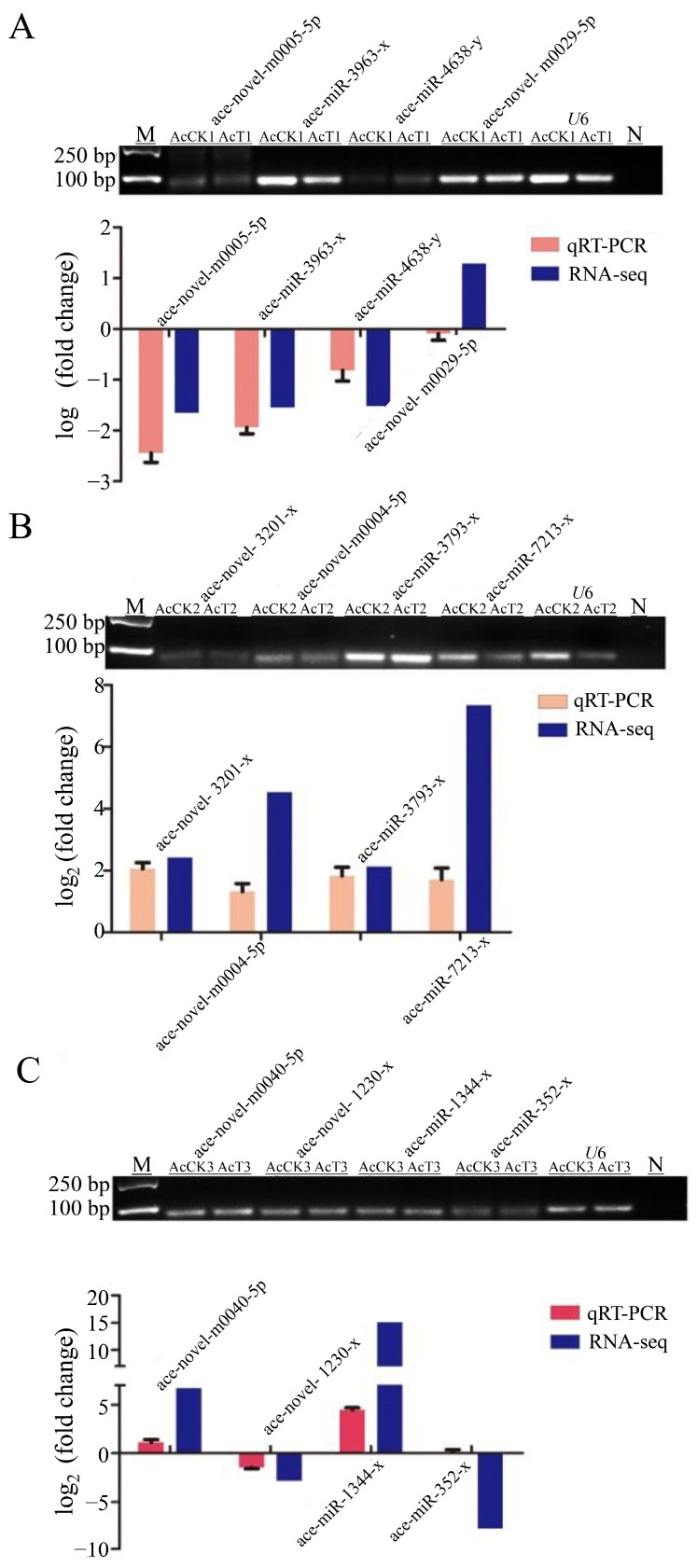
The trend of DEmiRNAs expression was verified by stem-loop RT-qPCR compared with RNA-Seq data. (**A**) The trend of DEmiRNAs expression of AcCK1 and AcT1 group (**B**) The trend of DEmiRNAs expression of AcCK2 and AcT2 group (**C**) The trend of DEmiRNAs expression of AcCK3 and AcT3 group. M: DNA marker; N: Negative controls.

**Figure 7 genes-16-00156-f007:**
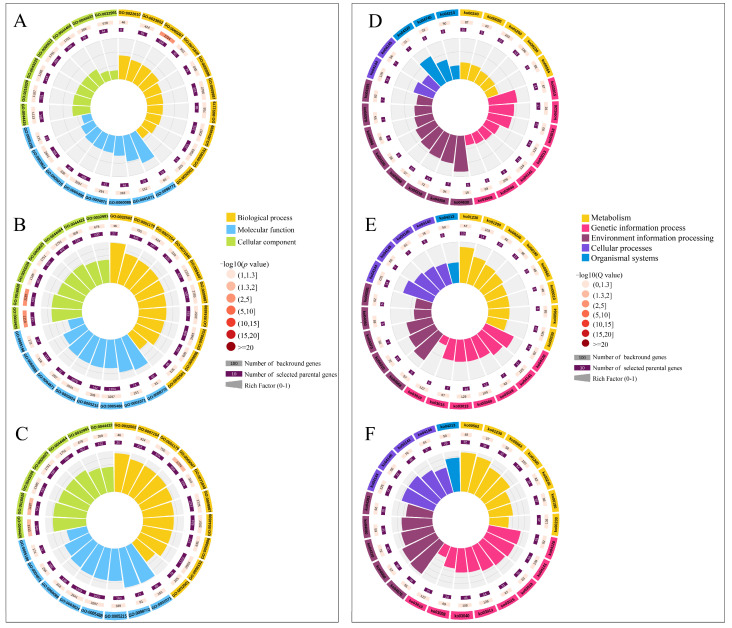
Loop graph of GO terms and KEGG pathways enrichment analysis of DEmiRNAs targeted mRNAs in AcCK vs. AcT group. (**A**) GO terms annotated by parental genes in AcCK1 vs. AcT1 group. (**B**) GO terms annotated by parental genes in AcCK2 vs. AcT2 group. (**C**) GO terms annotated by parental genes in AcCK2 vs. AcT2 group. (**D**) KEGG pathways enriched by parental genes in AcCK1 vs. AcT1 group. (**E**) KEGG pathways enriched by parental genes in AcCK2 vs. AcT2 group. (**F**) KEGG pathways enriched by parental genes in AcCK3 vs. AcT3 group.

**Figure 8 genes-16-00156-f008:**
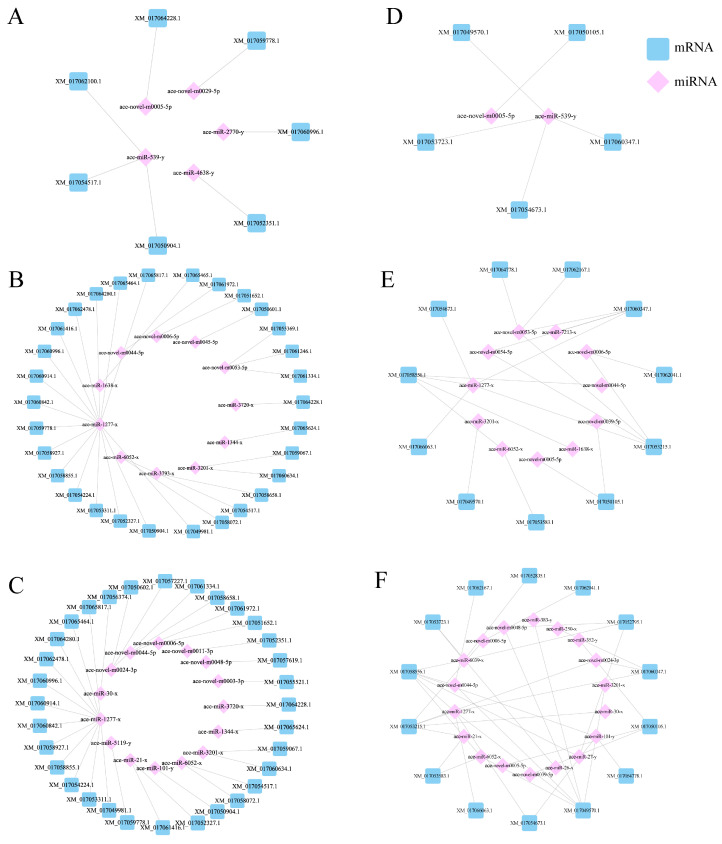
Construction of DEmiRNA-mRNA expression regulatory network based on JAK-STAT signaling pathway and phagosome. (**A**) DEmiRNA-mRNA expression regulatory network on JAK-STAT signaling pathway in AcCK1 vs. AcT1 group. (**B**) DEmiRNA-mRNA expression regulatory network on JAK-STAT signaling pathway in AcCK2 vs. AcT2 group. (**C**) DEmiRNA-mRNA expression regulatory network on JAK-STAT signaling pathway in AcCK3 vs. AcT3 group. (**D**) DEmiRNA-mRNA expression regulatory network on phagosome in AcCK1 vs. AcT1 group. (**E**) DEmiRNA-mRNA expression regulatory network on phagosome in AcCK2 vs. AcT2 group. (**F**) DEmiRNA-mRNA expression regulatory network on phagosome in AcCK3 vs. AcT3 group.

**Table 1 genes-16-00156-t001:** The name of the sequencing samples.

Age of Workers	4-Day-Old	5-Day-Old	6-Day-Old
Control	AcCK1-1 AcCK1-2 AcCK1-3	AcCK2-1 AcCK2-2 AcCK2-3	AcCK3-1 AcCK3-2 AcCK3-3
*A. apis*-infected	AcT1-1 AcT1-2 AcT1-3	AcT2-1 AcT2-2 AcT2-3	AcT3-1 AcT3-2 AcT3-3

**Table 2 genes-16-00156-t002:** Information about primers used in this study.

Primers	Sequence (5′-3′)
ace-novel-m0005-5p	Loop:CTCAACTGGTGTCGTGGAGTCGGCAATTCAGTTGAGACACCACGACACCACG F:ACACTCCAGCTGGGCGCGTGCAAGTT
ace-miR-4968-y	Loop:CTCAACTGGTGTCGTGGAGTCGGCAATTCAGTTGAGACACCACGTGCTGCTG F:ACACTCCAGCTGGGCAGCAGCAGCAG
ace-miR-3963-x	Loop:CTCAACTGGTGTCGTGGAGTCGGCAATTCAGTTGAGACACCACGCGGTGTCA F:ACACTCCAGCTGGGTATCCCACTCC
ace-miR-4638-y	Loop:CTCAACTGGTGTCGTGGAGTCGGCAATTCAGTTGAGACACCACGCGGCTGAG F:ACACTCCAGCTGGGCCTGGAAACGG
ace-novel-m0029-5p	Loop:CTCAACTGGTGTCGTGGAGTCGGCAATTCAGTTGAGACACCACGCACCGACT F:ACACTCCAGCTGGGTTGTTGCTATTATT
ace-miR-2770-y	Loop:CTCAACTGGTGTCGTGGAGTCGGCAATTCAGTTGAGACACCACGACCACTAC F:ACACTCCAGCTGGGGTTATCCCCGT
ace-miR-152-y	Loop:CTCAACTGGTGTCGTGGAGTCGGCAATTCAGTTGAGACACCACGCCAAGTTC F:ACACTCCAGCTGGGGTTATCCCCGT
ace-miR-1672-x	Loop:CTCAACTGGTGTCGTGGAGTCGGCAATTCAGTTGAGACACCACGTCCCTTCC F:ACACTCCAGCTGGGCGGTCAGGCCC
ace-miR-539-y	Loop:CTCAACTGGTGTCGTGGAGTCGGCAATTCAGTTGAGACACCACGAAAAAAGA F:ACACTCCAGCTGGGAAGTATAATT
ace-miR-7085-x	Loop:CTCAACTGGTGTCGTGGAGTCGGCAATTCAGTTGAGACACCACGTGCTGGCC F:ACACTCCAGCTGGGGGTGGGGGCC
ace-miR-9277-y	Loop:CTCAACTGGTGTCGTGGAGTCGGCAATTCAGTTGAGACACCACGAGTCGGTA F:ACACTCCAGCTGGGAGGTTCGAGTCC
ace-miR-1344-x	Loop:CTCAACTGGTGTCGTGGAGTCGGCAATTCAGTTGAGACACCACGCCGAGCAC F:ACACTCCAGCTGGGTGGGAAATGT
ace-miR-1777-x	Loop:CTCAACTGGTGTCGTGGAGTCGGCAATTCAGTTGAGACACCACGCGCCCCCG F:ACACTCCAGCTGGGCGCCCCCGAC
ace-novel-m0006-5p	Loop:CTCAACTGGTGTCGTGGAGTCGGCAATTCAGTTGAGACACCACGGCATGGTA F:ACACTCCAGCTGGGGCATGGTATATCTCA
ace-novel-m0016-5p	Loop:CTCAACTGGTGTCGTGGAGTCGGCAATTCAGTTGAGACACCACGTATACCCT F:ACACTCCAGCTGGGTATACCCTAGAA
ace-miR-143-y	Loop:CTCAACTGGTGTCGTGGAGTCGGCAATTCAGTTGAGACACCACGAGAGCTAC F:ACACTCCAGCTGGGAGAGCTACAGTG
ace-miR-21-x	Loop:CTCAACTGGTGTCGTGGAGTCGGCAATTCAGTTGAGACACCACGTCAACATC F:ACACTCCAGCTGGGTCAACATCAGTCTG
ace-miR-27-y	Loop:CTCAACTGGTGTCGTGGAGTCGGCAATTCAGTTGAGACACCACGGCAGAACT F:ACACTCCAGCTGGGGCAGAACTTAGCC
ace-miR-30-x	Loop:CTCAACTGGTGTCGTGGAGTCGGCAATTCAGTTGAGACACCACGAGCTTCCA F:ACACTCCAGCTGGGAGCTTCCAGTCGGGGA
ace-miR-342-y	Loop:CTCAACTGGTGTCGTGGAGTCGGCAATTCAGTTGAGACACCACGGGTGCGAA F:ACACTCCAGCTGGGGGTGCGAATTCT
ace-miR-352-y	Loop:CTCAACTGGTGTCGTGGAGTCGGCAATTCAGTTGAGACACCACGACTATACA F:ACACTCCAGCTGGGACTATACAACCT
ace-miR-4577-y	Loop:CTCAACTGGTGTCGTGGAGTCGGCAATTCAGTTGAGACACCACGCGTTCGCT F:ACACTCCAGCTGGGCGTTCGCTAT
ace-miR-6039-x	Loop:CTCAACTGGTGTCGTGGAGTCGGCAATTCAGTTGAGACACCACGACGTAAAC F:ACACTCCAGCTGGGACGTAAACTCACGC
ace-miR-9277-y	Loop:CTCAACTGGTGTCGTGGAGTCGGCAATTCAGTTGAGACACCACGAGTCGGTA F:ACACTCCAGCTGGGAGTCGGTAGGAC
ace-novel-m0024-3p	Loop:CTCAACTGGTGTCGTGGAGTCGGCAATTCAGTTGAGACACCACGAGGTAATA F:ACACTCCAGCTGGGAGGTAATAACTTGA
Primer-R	CTCAACTGGTGTCGTGGA

**Table 3 genes-16-00156-t003:** The number of Top 3 miRNAs and regulated mRNAs in DEmiRNAs in the guts of *A. c. cerana* larvae.

	Top3 miRNA	Number of mRNA
AcCK1 vs. AcT1 group	ace-miR-539-y	1464
	ace-miR-152-y	426
	ace-novel-m0029-5p	320
AcCK2 vs. AcT2 group	ace-miR-1277-x	6018
	ace-novel-m0053-5p	3037
	ace-novel-m0054-5p-	3037
AcCK3 vs. AcT3 group	ace-miR-1277-x	6018
	ace-novel-m0024-3p	1837
	ace-miR-6052-x	1474

## Data Availability

The original contributions presented in this study are included in the article. Further inquiries can be directed to the corresponding authors.

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
