# Peer review of "Transcriptomic Characterization of miRNAs in Apis cerana Larvae Responding to Ascosphaera apis Infection"

_genes, 2025, doi:10.3390/genes16020156_

Round 1
Reviewer 1 Report
Comments and Suggestions for Authors
Review report
This study provides valuable insights into the role of miRNAs in the immune response of A. cerana cerana larvae to A. apis infection, which causes chalkbrood disease. By employing small RNA sequencing (sRNA-seq) and regulatory network analyses, the manuscript offers a detailed examination of miRNA-mRNA interactions involved in immune responses. The results contribute to a deeper understanding of host-pathogen interactions and hold potential implications for improving honeybee health and mitigating the decline of bee colonies. However, while the study is well-structured and presents significant findings, some areas require clarification, and additional detail.
Comments for authors
1. The manuscript mentions that bee colonies were reared in an experimental apiary but does not provide critical details such as the number of colonies used, colony strength (e.g., number of frames of bees or brood), or the genetic diversity of the bees.
2. The health of the A. c. cerana colonies is stated to have been confirmed by PCR for A. apis-free status, but the methods for this verification are not described. Was this done for all colonies or only a subset? Details of the PCR protocols, primers, and controls used should be included.
3. The control group is described as being fed a diet without conidia, but there is no mention of how potential contamination was avoided during larval handling and feeding.
4. While the section mentions three biological replicates, the total number of larvae used per group and per time point is not specified.
Author Response
Reviewer1:
Comments to the Author
This study provides valuable insights into the role of miRNAs in the immune response of A. cerana cerana larvae to A. apis infection, which causes chalkbrood disease. By employing small RNA sequencing (sRNA-seq) and regulatory network analyses, the manuscript offers a detailed examination of miRNA-mRNA interactions involved in immune responses. The results contribute to a deeper understanding of host-pathogen interactions and hold potential implications for improving honeybee health and mitigating the decline of bee colonies. However, while the study is well-structured and presents significant findings, some areas require clarification, and additional detail.
Thank you very much for your comments concerning our manuscript. Those comments are all valuable and very helpful for revising and improving our manuscript, as well as the important guiding significance to our research. We have tried our best to improve the manuscript and have made a lot of changes which we hope meet with approval.
Comment 1: The manuscript mentions that bee colonies were reared in an experimental apiary but does not provide critical details such as the number of colonies used, colony strength (e.g., number of frames of bees or brood), or the genetic diversity of the bees.
Response 1: Thank you for your valuable advice. We have added to the key information on rearing colonies in experimental apiaries.
Original: A. c. cerana worker larvae were derived from strong colonies reared in the apiary of the College of Bee Science and Biomedicine, Fujian Agriculture and Forestry University, Fuzhou, China.
Revision (lines 107-109): A. c. cerana worker larvae were derived from three healthy brood comb of three different strong colonies (The new queen lays about 800 eggs a day, the number of workers fills the four nest frames) reared in the apiary of the College of Bee Science and Biomedicine, Fujian Agriculture and Forestry University, Fuzhou, China.
Comment 2: The health of the A. c. cerana colonies is stated to have been confirmed by PCR for A. apis-free status, but the methods for this verification are not described. Was this done for all colonies or only a subset? Details of the PCR protocols, primers, and controls used should be included.
Response 2: Accepted. We supplemented the detailed methods in Materials and Methods “Honey bee larvae and fungi”. In this study, we have performed PCR validation on samples of A. c. cerana worker larvae from 3 colonies to ensure that bee larvae were free of A. apis infection.
Revision (lines 114-126): We performed PCR validation on A. c. cerana worker samples from colonies to ensure that the bee larvae were not infected with A. apis. The intestines of 20 larvae were dissected with tweezers sterilized with 70% alcohol and placed in sterile tubes (n=20). The total RNA of the intestinal tracts of bee larvae was extracted with the gRNAiso plus kit (TaKaRa, Dalian, China), and then reverse transcribed into cDNA. RT-PCR system (20 μL): cDNA 1 μL, 2×Hieff®PCR Master Mix 10 μL (Yeasen, Shanghai, China), Primer-F (5'-TCTGGCGGCCGGTTAAAGGCTTC-3') 1 μL, Primer-R (5'-GTTTCAAGACGGGCCACAAAC-3') 1 μL, ddH2O 7 μL. PCR procedure: pre-denaturation at 95°C for 5 min; denaturation at 95°C for 40 s, annealing at 55°C for 30 s, extension at 72°C for 40 s, 34 cycles; extending at 72°C for 10 min. The products were detected by 1.5% agarose gel electrophoresis and gel imager (Peiqing, China). Colonies of worker bee samples without amplification bands were selected for future research.
Comment 3: The control group is described as being fed a diet without conidia, but there is no mention of how potential contamination was avoided during larval handling and feeding.
Response 3: We have added methods to prevent the control group larvae from becoming infected during larval handling and feeding.
Original: The 3-day-old larvae in the control group were fed a diet without conidia 20 μL, and then all larvae were fed a diet without conidia.
Revision (lines 142-146): The 3-day-old larvae in the control group (n=48) were fed a diet without conidia 20 μL with a 70 % alcohol disinfected shifting needle, and then all larvae were fed a diet without conidia on the ultra-clean workbench. Control group and A. apis infected group were reared in different sterile incubators to avoid potential infection.
Comment 4: While the section mentions three biological replicates, the total number of larvae used per group and per time point is not specified.
Response 4: Thank you for your valuable advice. We have supplemented the total number of larvae used in each group and at each time point.
Original: The 2-day-old larvae were transplanted into a sterile 48-well cell culture plate with preset 40 μL feed (preheated at 35°C) with a shifting needle, and cultured in a constant temperature and humidity box at 35 ± 0.5°C and 90% relative humidity. The feed was changed every 24 h. Larvae in the treatment group were fed a premixed diet containing 20 μL of conidia at 3 days of age, with conidia concentration of 1×107 spores/mL. The 3-day-old larvae in the control group were fed a diet without conidia 20 μL, and then all larvae were fed a diet without conidia. The gut of larvae in the control group (AcCK) and the Ascosphaera apis infected group (AcT) were collected on day 1, day 2 and day 3 after infection, respectively.
Revision (lines 137-149): The 2-day-old larvae (n=96) were transplanted into two sterile 48-well cell culture plate with preset 40 μL feed (preheated at 35°C) with shifting needle, and cultured in a constant temperature and humidity box at 35 ± 0.5°C and 90% relative humidity. The feed was changed every 24 h. Larvae in the A. apis infected group (n=48) were fed a premixed diet containing 20 μL of conidia at 3 days of age, with conidia concentration of 1×107 spores/mL. The 3-day-old larvae in the control group (n=48) were fed a diet without conidia 20 μL with a 70 % alcohol disinfected shifting needle, and then all larvae were fed a diet without conidia on the ultra-clean workbench. Control group and A. apis infected group were reared in different sterile incubators to avoid potential infection. The gut of larvae in the control group (AcCK) and the A. apis infected group (AcT) were collected on day 1, day 2 and day 3 after infection, respectively. Three gut samples were mixed as one biological sample, and the experiment used three independent biological samples.

Reviewer 2 Report
Comments and Suggestions for Authors
This is an interesting article reporting on the transcriptomic characterization of miRNAs in Apis cerana larvae in response to Ascosphaera apis infection. The authors found that miRNA is involved in the immune response of honeybees to A. apis infection by regulating energy metabolism, cellular immunity and humoral immunity. The results obtained provide a basis for the regulation of miRNAs in honeybee larvae in response to A. apis infection, and provide new insights into host-pathogen interactions.
Minor changes
Line 67: delete full stop before parenthesis
Line 117: add full stop after workers
Line 149-158: this information should be better presented as a table
Line 166: A. mellifera, use Italics
An editing of the whole manuscript is suggested.
Author Response
Reviewer2:
Comments to the Author
This is an interesting article reporting on the transcriptomic characterization of miRNAs in Apis cerana larvae in response to Ascosphaera apis infection. The authors found that miRNA is involved in the immune response of honeybees to A. apis infection by regulating energy metabolism, cellular immunity and humoral immunity. The results obtained provide a basis for the regulation of miRNAs in honeybee larvae in response to A. apis infection, and provide new insights into host-pathogen interactions.
Thank you very much for your comments concerning our manuscript. Those comments are all valuable and very helpful for revising and improving our manuscript, as well as the important guiding significance to our research. We have tried our best to improve the manuscript and have made a lot of changes which we hope meet with approval.
Minor changes:
Comment 1: Line 67: delete full stop before parenthesis
Response 1: We have deleted full stop before parenthesis and checked the punctuation in the manuscript.
Comment 2: Line 117: add full stop after workers
Response 2: We have checked and modified the punctuation in the manuscript.
Comment 3: Line 149-158: this information should be better presented as a table
Response 3: Accepted. We present this information in table form.
Original: Ultimately, the 18 cDNA libraries were sequenced on Illumina sequencing platform (HiSeqTM 4000) using the single-end technology by GENE DENOVO Biotechnology Co. (Guangzhou, China). The libraries were as follows: AcCK 1-1, AcCK 1-2 and AcCK 1-3 as replicate libraries for normal guts of 4 day-old-workers with sucrose solution; AcT1-1, AcT1-2 and AcT1-3 as replicate libraries for guts of 4 day-old-workers at 1 day post inoculation (dpi) with sucrose solution containing A. apis spores; AcCK2-1, AcCK2-2 and AcCK2-3 as replicate libraries for normal guts of 5 day-old-workers with sucrose solution; AcT2-1, AcT2-2 and AcT2-3 as replicate libraries for guts of 5-day-old workers at 2 dpi with sucrose solution containing A. apis spores. AcCK3-1, AcCK3-2 and AcCK3-3 as replicate libraries for normal guts of 6 day-old-workers with sucrose solution; AcT3-1, AcT3-2 and AcT3-3 as replicate libraries for guts of 3-day-old workers at 3 dpi with sucrose solution containing A. apis spores. The data measured in this study have been uploaded to the National Center for Biotechnology Information (NCBISRA) database, BioProject number: PRJNA395108
Revision (lines 174-175): Ultimately, the 18 cDNA libraries were sequenced on Illumina sequencing platform (HiSeqTM 4000) using the single-end technology by GENE DENOVO Biotechnology Co. (Guangzhou, China). The names of these sequencing samples are shown in Table 1. The data measured in this study have been uploaded to the National Center for Biotechnology Information (NCBISRA) database, BioProject number: PRJNA395108.
Table 1. The name of the sequencing samples
Age of workers |
4-day-old |
5-day-old |
6-day-old |
Control |
AcCK1-1/2/3 |
AcCK2-1/2/3 |
AcCK3-1/2/3 |
A. apis infected |
AcT1-1/2/3 |
AcT2-1/2/3 |
AcT3-1/2/3 |
Comment 4: Line 166: A. mellifera, use Italics An editing of the whole manuscript is suggested.
Response 4(Line 185): We have changed the format of "A. mellifera" to italics in the full text.

Reviewer 3 Report
Comments and Suggestions for Authors
The study presents a first description of miRNAs in an Apis species in relation to a fungal pathogen infection. Methodologically, the study is correctly designed and presented in a very good manner. However there are a few parts that need revision. Particularly, apart from a few minor comments that follow, the discussion lacks a comparison with other Apis species miRNAs or at least if no such data exist in the literature, a comparison with corresponding physiological studies with other Apis species. As it is, the discussion is only deals with miRNAs in mode organisms or mammals.
Additionally the reference in the abstract of host energy metabolism, cellular immunity and humoral immunity need better support in the discussion. How these traits were measured / evaluated?
Thus, I recommend a deep revision of the discussion, as well as the implementation of some minor corrections that follow
lines 29-30: The decline in the number of honeybees poses….of global agriculture. This is irrelevant with the present study and mostly not a proven fact, but just a “trendy” statement. I recommend to delete it
lines 87-90: How parasite miRNA regulate the host response? Please explain more in detail
line 102: please define “strong” colonies
lines 113-114: (as verified by PCR). This has to be in detail described in Materials and Methods
Lines 170-171: this is repeated above, please delete
Author Response
Reviewer3:
The study presents a first description of miRNAs in an Apis species in relation to a fungal pathogen infection. Methodologically, the study is correctly designed and presented in a very good manner. However, there are a few parts that need revision. Particularly, apart from a few minor comments that follow, the discussion lacks a comparison with other Apis species miRNAs or at least if no such data exist in the literature, a comparison with corresponding physiological studies with other Apis species. As it is, the discussion is only deals with miRNAs in mode organisms or mammals.
Response: Thank you for your valuable advice. In addition to model organisms or mammals, we supplemented the discussion of miRNA of other Apis species. The modifications are as follows:
Original: Over the past decade, miRNA has become a global research hotspot and has attracted more and more attention (Seyhan et al., 2024). The vast majority of miRNA research is carried out in animals, plants and insects, especially in model species such as Homo sapiens (Ghosh et al., 2024) and Arabidopsis thaliana (Pegler et al., 2024) and Drosophila (Jang et al., 2024). Based on sRNA-seq, this study analyzed the miRNA differences in the larval intestine of A. c. cerana at 1 dpi, 2 dpi and 3 dpi. The results showed that a total of 537 miRNAs were detected and 73 DEmiRNAs were obtained. DEmiRNAs targeted 2778 mRNAs, and GO entries were enriched to binding, cellular process, catalytic activity, metabolic process and single-organism process (Figure 6A). The main pathways enriched by KEGG were endocytosis, ubiquitin-mediated proteolysis, RNA transport, JAK-STAT immune-related signaling pathways and phagosome (Figure 6), and the miRNA-mRNA regulatory network enriched by JAK-STAT signaling pathway was established (Figure 6D). The results of this study provide a rich genetic resource for the function of miRNAs in the response of A.c.cerana larvae to A. apis infection, and provide new insights into host-pathogen interactions.
Revision (Lines 404-415): Over the past decade, miRNA has become a global research hotspot and has attracted more and more attention (Seyhan et al., 2024). The vast majority of miRNA research is carried out in animals, plants and insects, especially in model species such as Homo sapiens (Ghosh et al., 2024) and Arabidopsis thaliana (Pegler et al., 2024) and Drosophila (Jang et al., 2024). The research on miRNA in bees has also received increasing attention. Bumblebees (Bombus terrestris) infected with slow bee paralysis virus (SBPV) showed increased expression of two miRNAs, dicer-1 and ago-1. There were 17 and 12 differentially expressed miRNAs in SBPV and Israeli acute paralysis virus (IAPV) infection, respectively (Niu et al., 2017). Female alfalfa leafcutting bees (Megachile rotundata) adjust the miRNA deposition in response to seasonal changes (Hagadorn et al., 2023). Ame-miR-980-3p is involved in A. mellifera autophagy mediated midgut remodeling by targeting Atg2B (Chen et al., 2023). Ame-miR-519 influences the level of juvenile hormone through Eth, thereby regulating the transition from larva to pupa of A. mellifera workers (Dong et al., 2024). ame-miR-34 modulates larval body weight and immune response of A. mellifera workers to invasion by A. apis (Wu et al., 2023). The function of miRNAs is relatively conserved in both A. c. cerana and A. mellifera.
In this study, based on sRNA-seq, this study analyzed the miRNA differences in the larval intestine of A. c. cerana at 1 dpi, 2 dpi and 3 dpi. The results showed that a total of 537 miRNAs were detected and 73 DEmiRNAs were obtained. DEmiRNAs targeted 2778 mRNAs, and GO entries were enriched to binding, cellular process, catalytic activity, metabolic process and single-organism process (Figure 6A). The main pathways enriched by KEGG were endocytosis, ubiquitin-mediated proteolysis, RNA transport, JAK-STAT immune-related signaling pathways and phagosome (Figure 6), and the miRNA-mRNA regulatory network enriched by JAK-STAT signaling pathway was established (Figure 6D). The results of this study provide a rich genetic resource for the function of miRNAs in the response of A.c.cerana larvae to A. apis infection, and provide new insights into host-pathogen interactions.
Original: A single miRNA simultaneously regulates multiple target mRNAs to inhibit their expression, and vice versa (Xie et al., 2016). We observed that the DEmiRNA with the largest number of target genes regulated by the gut of A. c. cerana larvae in response to A. apis infection were ace-miR-539-y and ace-miR-1277-x, suggesting that ace-miR-539-y and ace-miR-1277-x may play an important regulatory role in the response of A. c. cerana larvae to A. apis infection (Figure 7). As an important regulator of insect immunity, miRNAs directly target immune-related genes (Monsanto-Hearne et al., 2019).
Revision (Lines 443-447): A single miRNA simultaneously regulates multiple target mRNAs to inhibit their expression, and vice versa (Xie et al., 2016). We observed that the DEmiRNA with the largest number of target genes regulated by the gut of A. c. cerana larvae in response to A. apis infection were ace-miR-539-y and ace-miR-1277-x, suggesting that ace-miR-539-y and ace-miR-1277-x may play an important regulatory role in the response of A. c. cerana larvae to A. apis infection (Figure 7). As combined with the previous studies which found that 165 differentially expressed miRNAs of N. ceranae as a whole targeted 11,711 mRNAs in the midgut of A. mellifera workers (Fan et al., 2021). We suggested that miRNA target multiple mRNA, taking part in the metabolism and immunity process of two Apis species of workers. As an important regulator of insect immunity, miRNAs directly target immune-related genes (Monsanto-Hearne et al., 2019).
References:
- Niu J, Meeus I, De Coninck DI, Deforce D, Etebari K, Asgari S, Smagghe G. Infections of virulent and avirulent viruses differentially influenced the expression of dicer-1, ago-1, and microRNAs in Bombus terrestris. Sci Rep. 2017 Apr 4;7:45620. doi: 10.1038/srep45620.
- Hagadorn MA, Hunter FK, DeLory T, Johnson MM, Pitts-Singer TL, Kapheim KM. Maternal body condition and season influence RNA deposition in the oocytes of alfalfa leafcutting bees (Megachile rotundata). Front Genet. 2023 Jan 4;13:1064332. doi: 10.3389/fgene.2022.1064332.
- Wu Y, Guo Y, Fan X, Zhao H, Zhang Y, Guo S, Jing X, Liu Z, Feng P, Liu X, Zou P, Li Q, Na Z, Zhang K, Chen D, Guo R. ame-miR-34 Modulates the Larval Body Weight and Immune Response of Apis mellifera Workers to Ascosphara apis Invasion. Int J Mol Sci. 2023 Jan 7;24(2):1214. doi: 10.3390/ijms24021214.
- Chen WF, Chi XP, Song HY, Wang HF, Wang Y, Liu ZG, Xu BH. Ame-miR-980-3p participates in autophagy-mediated midgut remodelling in Apis mellifera via targeting Atg2B. Insect Mol Biol. 2023 Dec;32(6):748-760. doi: 10.1111/imb.12869.
- Dong S, Li K, Zang H, Song Y, Kang J, Chen Y, Du L, Wang N, Chen D, Luo Q, Yan T, Guo R, Qiu J. ame-miR-5119-Eth axis modulates larval-pupal transition of western honeybee worker. Front Physiol. 2024 Sep 27;15:1475306. doi: 10.3389/fphys.2024.1475306.
- Fan XX, Du Y,Zhang WD,Wang J, Jiang HB, Fan YC, Feng RR, Wan JQ, Zhou ZY, Xiong CL, Zheng YZ, Chen DF, Guo R.Omics analysis of Nosema ceranae miRNAs involved in gene expression regulation in the midgut of Apis mellifera ligustica workers and their regulatory networks[J]. Acta Entom.Sin., 2021, 64 (02): 187-204.
Additionally, the reference in the abstract of host energy metabolism, cellular immunity and humoral immunity needs better support in the discussion. How these traits were measured / evaluated?
Response: Accepted. The causes of changes of host energy metabolism, cellular immunity and humoral immunity caused by A. apis infection were further explained in the Discussion section. The modifications are as follows:
Revision (Lines 460-475): The release, transfer, storage and utilization of energy in the metabolic process play an indispensable role in insect immune response (Bland, 2023). The invasion of pathogenic micro-organisms not only activates cellular immunity, including endocytosis, autophagy and phagosome, but also activates humoral immunity, such as JAK-STAT, Toll and Imd signaling pathways (Sun et al., 2024; Negri et al., 2019; McMenamin, 2018). Wang et al. (2021) analyzed that the 20E-HaEcR-HaUSP complex of Helicoverpa armigera stimulated the expression of HaCTL1 gene in hemolymph, thereby increasing the synthesis and secretion of HaCTL1 protein, promoting cell phagocytosis and encapsulation, and resisting the in-vasion of Ovomermis sinensis. Lourenço et al. (2013) detected that the expression of an-timicrobial peptide gene Abaecin in A.mellifera was regulated by both Toll and IMD signaling pathways through bioinformatics analysis, and the expression of hymenop-taecin and defensin1 was regulated by IMD and Toll signaling pathways, respectively.
References:
- Bland ML. Regulating metabolism to shape immune function: Lessons from Drosophila. Semin Cell Dev Biol. 2023 Mar 30;138:128-141. doi: 10.1016/j.semcdb.2022.04.002.
- Negri P, Villalobos E, Szawarski N, Damiani N, Gende L, Garrido M, Maggi M, Quintana S, Lamattina L, Eguaras M. Towards Precision Nutrition: A Novel Concept Linking Phytochemicals, Immune Response and Honey Bee Health. Insects. 2019 Nov 12;10(11):401. doi: 10.3390/insects10110401.
- Wang G J,Wang W W, Liu Y, Chai LQ,Wang GX, Liu XS, Wang YF, Wang JL.Steroid hormone 20-hydroxyecdysone promotes CTL1-mediated cellular immunity in Helicoverpa armigera [J].Insect Science,2021,28(5):1399-1413.
- Lourenço AP, Guidugli-Lazzarini KR, Freitas FC, Bitondi, M M, Simões ZL. Bacterial infection activates the immune system response and dysregulates microRNA expression in honey bees[J]. Insect Biochemistry and Molecular Biology, 2013, 43(5): 474-482.
Thus, I recommend a deep revision of the discussion, as well as the implementation of some minor corrections that follow.
Thank you very much for your comments concerning our manuscript. Those comments are all valuable and very helpful for revising and improving our manuscript, as well as the important guiding significance to our research. We have tried our best to improve the manuscript and have made a lot of changes which we hope meet with approval.
Minor corrections
Comment 1: lines 29-30: The decline in the number of honeybees poses….of global agriculture. This is irrelevant with the present study and mostly not a proven fact, but just a “trendy” statement. I recommend to delete it
Response 1: Accepted. We have deleted this sentence.
Comment 2: lines 87-90: How parasite miRNA regulate the host response? Please explain more in detail.
Response 2: Accepted. We have added detailed description how parasite miRNA regulates the host response in the Introduction.
Original: It was found that bee genes involved in apoptosis and innate immune responses are regulated by both host and parasite miRNAs through the analysis of small RNA sequencing and expression profile of miRNAs and their target genes of hosts and pathogens during Nosema ceranae infection (Evans et al. 2018).
Revision (Lines 84-95): Studies have shown that viruses or parasites produce host-like miRNAs that regulate host gene expression (Diggins and Hancock, 2023). Over the past decade, miRNAs of Leishmania (L. donovani) have been shown to be involved in the pathogenesis of leishmaniasis (Paul et al., 2020). miR-361-3p and miR-140-3p were significantly overexpressed in cutaneous leishmaniasis lesions generated by L. braziliensis infection as compared to normal skin samples targeting genes involved in worsening of tissue damage (Lago et al., 2018). Evans et al. (2018) found that 918 honeybee genes involved in biological processes such as apoptosis and the innate immune response were broadly negatively regulated by five parasite miRNAs by analyzing the expression profiles of host and pathogen miRNAs and their target genes during honeybee infections. Fan et al. (2022) found that a total of 343 down-regulated mRNAs in A. mellifera were putative targets for 121 up-regulated miRNAs in N.cerana, which were mainly enriched in 217 KEGG pathways, including JAK-STAT signaling pathway, endocytosis, and lysosomes.
- Fan, X.X., Zhang, W.D., Zhang, K.Y., Zhang, J.X., Long, Q., Wu, Y., Zhang, K.H.; Zhu,L.R.; Chen DF, Guo, R. In-depth investigation of microRNA-mediated cross-kingdom regulation between Asian honey bee and microsporidian. Front. Microbiol., 2022, 13, 1003294.
- Diggins NL, Hancock MH. Viral miRNA regulation of host gene expression. Semin Cell Dev Biol. 2023 Sep 15;146:2-19. doi: 10.1016/j.semcdb.2022.11.007. Epub 2022 Nov 30.
- Lago TS, Silva JA, Lago EL, Carvalho EM, Zanette DL, Castellucci LC. The miRNA 361-3p, a Regulator of GZMB and TNF Is Associated With Therapeutic Failure and Longer Time Healing of Cutaneous Leishmaniasis Caused by L. (viannia) braziliensis. Front Immunol. 2018 Nov 14;9:2621. doi: 10.3389/fimmu.2018.02621.
- Paul S, Ruiz-Manriquez LM, Serrano-Cano FI, Estrada-Meza C, Solorio-Diaz KA, Srivastava A. Human microRNAs in host-parasite interaction: a review. 3 Biotech. 2020 Dec;10(12):510. doi: 10.1007/s13205-020-02498-6. Epub 2020 Nov 5.
Comment 3: line 102: please define “strong” colonies
Response 3: Thank you for your valuable advice. We have supplemented the definition of "strong colonies".
Original: A. c. cerana worker larvae were derived from strong colonies reared in the apiary of the College of Bee Science and Biomedicine, Fujian Agriculture and Forestry University, Fuzhou, China.
Revision (lines 107-109): A. c. cerana worker larvae were derived from three healthy brood comb of three different strong colonies (The new queen lays about 800 eggs a day; The number of worker bees fills the four nest frames) reared in the apiary of the College of Bee Science and Biomedicine, Fujian Agriculture and Forestry University, Fuzhou, China.
Comment 4: lines 113-114: (as verified by PCR). This has to be in detail described in Materials and Methods
Response 4: Accepted. We supplemented the detailed methods in Materials and Methods “Honey bee larvae and fungi”. In this study, we have performed PCR validation on samples of A. c. cerana worker larvae from 3 colonies to ensure that bee larvae were free of A. apis infection.
Revision (lines 114-126): We performed PCR validation on A. c. cerana worker samples from colonies to ensure that the bee larvae were not infected with A. apis. The intestines of 20 larvae were dissected with tweezers sterilized with 70% alcohol and placed in sterile tubes (n=20). The total RNA of the intestinal tracts of bee larvae was extracted with the gRNAiso plus kit (TaKaRa, Dalian), and then reverse transcribed into cDNA. RT-PCR system (20 μL): cDNA 1 μL, 2×Hieff®PCR Master Mix 10 μL (Yeasen, Shanghai, China), Primer-F (5'-TCTGGCGGCCGGTTAAAGGCTTC-3') 1 μL, Primer-R (5'-GTTTCAAGACGGGCCACAAAC-3') 1 μL, ddH2O 7 μL. PCR procedure: pre-denaturation at 95°C for 5 min; denaturation at 95°C for 40 s, annealing at 55°C for 30 s, extension at 72°C for 40 s, 34 cycles; extending at 72°C for 10 min. The products were detected by 1.5% agarose gel electrophoresis and gel imager (Peiqing, China). Colonies of worker bee samples without amplification bands were selected for future research.
Comment 5: Lines 170-171: this is repeated above, please delete.
Response 5: Accepted. We have changed this sentence.
Original: Total RNA was extracted from the samples by Trizol method. Fragments of 18-30 nt were selected for agarose gel electrophoresis, and then 3' RNA dapter were ligatied.
Revision (lines 194-195): The RNA fragments of 18-30 nt were selected for agarose gel electrophoresis, and then 3' RNA dapter were ligatied.

Round 2
Reviewer 1 Report
Comments and Suggestions for Authors
The authors have answered all my questions. I have no additional comments. I recommend that the MS be accepted for publication.